# Internal Carotid Injury during Skull Base Surgery—Case Report and a Review of the Literature

**DOI:** 10.3390/brainsci12091254

**Published:** 2022-09-16

**Authors:** Petr Matoušek, Tomáš Krejčí, Eva Misiorzová, Radim Lipina, Václav Procházka, Jakub Lubojacký, Lenka Čábalová, Pavel Komínek

**Affiliations:** 1Department of Otorhinolaryngology and Head and Neck Surgery, University Hospital Ostrava, 708 52 Ostrava, Czech Republic; 2Department of Craniofacial Surgery, Faculty of Medicine, University of Ostrava, 701 03 Ostrava, Czech Republic; 3Department of Neurosurgery, University Hospital Ostrava, 708 52 Ostrava, Czech Republic; 4Department of Clinical Neurosciences, Faculty of Medicine, University of Ostrava, 701 03 Ostrava, Czech Republic; 5Department of Radiology, University Hospital Ostrava, 708 52 Ostrava, Czech Republic; 6Department of Imaging Methods, Faculty of Medicine, University of Ostrava, 701 03 Ostrava, Czech Republic

**Keywords:** internal carotid artery injury, iatrogenic injury, transnasal skull base surgery, angiography, endovascular treatment

## Abstract

Iatrogenic injury of the internal carotid artery (ICA) is a rare, and probably underreported, complication of transnasal endoscopic skull base surgery. Although treatment algorithms have been suggested, there is no definite consensus or guideline for the management of this severe complication. We describe a case of ICA injury that occurred during a transsphenoidal biopsy of a tumor in the cavernous sinus and we present a treatment algorithm for managing this complication. We reviewed the articles published from 1998 to 2021, reporting on major vascular injury during transnasal endoscopic skull base surgery and endonasal endoscopic surgery, and we compare the methods and results of ICA injury management reported in the literature with the presented case. The most promising treatment for ICA injury might be packing with a muscle graft initially, then performing an endovascular intervention.

## 1. Introduction

Iatrogenic injury of the internal carotid artery (ICA) is a rare complication of transnasal endoscopic skull base surgery, which can potentially lead to permanent or even fatal sequelae. The lethality of this injury is approximately 10% [1]. According to the literature, the incidence of iatrogenic ICA injuries is relatively low, with rates of 0.12–1.1% in transsphenoidal approaches and 4–9% in extended endoscopic approaches for treating skull base pathology [2,3,4]. However, the actual incidences are probably substantially higher [5]. According to a survey by Rowan et al., more than 20% of skull base surgeons have caused an ICA injury during their careers; therefore, it is assumed that this complication has been underreported in the scientific literature [5]. Good cooperation of a multidisciplinary team consisting of otorhinolaryngologists, neurosurgeons, anesthesiologists, interventional radiologists, neurologists, and physical therapists is fundamental for optimal results [6].

Although treatment algorithms have been suggested and training models have been created, there is no definite consensus or guideline for the management of this severe complication [3,7,8,9]. Most ICA injuries reported in the literature occurred during standard transsphenoidal surgeries for intrasellar pathologies with a binostril approach or in extended approaches. Here, we aimed to describe a case of iatrogenic ICA injury in a patient with a metastatic tumor involving the right cavernous sinus and encircling the right ICA, in which a mononostril approach was used for the biopsy of the tumor. An algorithm of management of this complication is proposed.

For the literature review, we searched the PubMed, MEDLINE, Scopus, and Cochrane databases; a combination of keywords, “internal carotid artery injury, iatrogenic injury, transnasal skull base surgery, angiography, endovascular treatment”, was used in the search strategy.

## 2. Case Description

A 66-year-old female presented with hypesthesia and pain in the right half of the face. A tumor was suspected. She had a history of metastatic mammary ductal carcinoma, which was treated 5 years prior to the visit. Magnetic resonance imaging (MRI) showed a right cavernous sinus tumor, which extended to the cavum Meckeli and encircled the right ICA cavernous segment (Figure 1a). A transsphenoidal endoscopic biopsy was performed for a pathological workup.

The interdisciplinary surgical team consisted of an otorhinolaryngologist and a neurosurgeon. The surgery was performed under general anesthesia and under optical image guidance control, with a fusion of MRI and computed tomography (CT). The right sphenoid sinus was opened with the mononostril endoscopic approach to visualize the tumor. The tumor had expanded into the clival recess. The position of the cavernous segment of ICA was repeatedly verified with image guidance—the vessel lumen was visualized at approximately 1 cm from the location of the intended biopsy, in the clival recess.

Immediately after the biopsy was performed with Blakesley forceps, profuse arterial bleeding appeared. Initially, the bleeding was managed with thrombin foam and gauze strips. Simultaneously, a second surgical team harvested a muscle graft (approximately 15 × 15 × 5 mm) from the rectus abdominis muscle from the periumbilical area. At that time, an interventional radiologist was contacted. It was impossible to visualize the exact origin of bleeding because the tumor had infiltrated the ICA. Therefore, the right sphenoid sinus was packed with a crushed muscle graft, and its position was secured with two polyvinyl acetate packing strips (Merocel, Medtronic Xomed Surgical Products, Jacksonville, FL, USA). This treatment stopped the bleeding and enabled patient transport to the angiography suite.

The estimated blood loss was 1.5–2 L in 5 min. Crystalloids and red blood cell transfusions were administered by the anesthesiology team. Despite these measures, the patient was in critical condition. Digital subtraction angiography (DSA) revealed a right ICA occlusion, though there was good collateral flow through the anterior and posterior communicating arteries (Figure 1c,d). Throughout the DSA examination, the patient was hemodynamically unstable, with bradycardia progressing to asystole. Extensive cardiopulmonary resuscitation (CPR) was performed for approximately 4–5 min. Due to the right ICA occlusion and good collateral flow through the anterior communicating artery, no endovascular intervention was indicated.

A subsequent brain CT was performed immediately after the DSA. The CT revealed a small subarachnoidal hematoma adjacent to the right cavernous sinus, and, overall, both the white and gray matter had dedifferentiated as a result of prolonged hypotension and CPR (Figure 1b).

After surgery, the patient was treated in the neurosurgical intensive care unit. Repeated brain CT scans showed extensive supra- and infratentorial ischemia, which led to malignant brain edema. Five days after the surgery, a patient exam showed a reactive mydriasis and the absence of brainstem reflexes. The patient died 6 days after the surgery. A histopathologic examination of the tissue biopsy confirmed a mammary ductal carcinoma metastasis in the right cavernous sinus.

## 3. Discussion

According to the literature, an ICA injury during skull base surgery is a rare, and probably underreported, complication. Although treatment algorithms were proposed by Gardner et al., Kassir et al., and Hamour et al., currently, there is no universal guideline for the management of this complication [7,10,11]. However, it is important to be prepared for it and to agree on an algorithm for managing ICA injuries that is feasible in the specific settings of a particular institution. The algorithm followed in our department is described in Figure 2. For the best results in minimizing morbidity and mortality, a fundamental feature of the algorithm is close cooperation within a broad interdisciplinary team, which comprises otorhinolaryngologists, neurosurgeons, anesthesiologists, interventional radiologists, neurologists, and rehabilitation therapists [6].

### 3.1. Epidemiology

In the last decade, transnasal endoscopic skull base surgery has become the treatment of choice for an increasing number of different pathologies. Its leading role in pituitary surgery is currently indisputable. The number of extended approaches to different skull base pathologies is also dramatically increasing. In absolute numbers, surgeries performed with the transsphenoidal approach caused more ICA injuries than surgeries performed with extended approaches. However, during extended approaches, it is often necessary to resect the bony ICA canal or manipulate the ICA. Therefore, the risk of ICA injury is several times higher (5–9%) for extended approaches, compared to transsphenoidal approaches (0.5–1.1%) [3,12,13,14].

The rate of left ICA injuries is higher than the rate of right ICA injuries (ratio = 1.3:1). This difference could be explained by the right-hand dominance of most surgeons. Thus, during surgery, right-handed surgeons tend to put higher pressure on the left lateral wall than on the right wall, according to Chin et al. [1].

The presented case study was the only ICA injury that occurred during a transnasal endoscopic surgery in our institution between 2010–2019. During that time, surgeons at our institution performed 420 endoscopic skull base surgeries, 253 transsphenoidal surgeries, and 167 extended approaches. Thus, our observed incidence (0.23%) was lower than the incidences reported in the literature.

### 3.2. Anatomy and Risk Factors

The ICA cavernous segment is most prone to injury in transnasal approaches to the skull base and in endoscopic sinus surgery [3,15]. The C2 (petrous) and C3 (lacerum) segments might also be injured during extended approaches [16,17]. Cavernous segment injuries occur most frequently, due to the higher frequency of transsphenoidal approaches and the high incidence of lateral sphenoid sinus wall dehiscence. According to cadaver studies, lateral sphenoid sinus wall dehiscence occurs in 4–22% of cases, and the wall is less than 0.5 mm thick in 88% of cases [13,18]. Moreover, in up to 10% of cases, the ICA canal courses within 4 mm of the midline [6]. Insertion of the intersphenoidal septum into the bony ICA canal was anatomically unfavorable in 16.3% of cases, according to Park et al., while, according to Dziedzic et al., 49% of sphenoid sinuses have at least one septum exhibiting involvement with the ICA protuberance [19,20]. During a resection, twisting and pushing movements can cause a bone fragment to injure the ICA wall. Therefore, the use of a diamond drill or cutting instruments is recommended [6,19,21].

The risk of ICA injury in transnasal endoscopic approaches increases in revision surgery, after skull base radiotherapy, or when tumors are in direct contact with the ICA [1]. In the present case study, the right ICA was infiltrated by a mammary carcinoma, and the ICA wall was injured during the tumor biopsy procedure.

### 3.3. Preoperative Imaging and Intraoperative Image Guidance

There is no clear consensus about the best preoperative imaging method to use in skull base surgery. In most studies on ICA injuries, the only imaging examination performed was a head CT; an MRI was the second most frequently used method. Only a few authors used CT angiography prior to surgery [1]. According to Kassir et al., both MRI and CT angiography should be performed prior to surgery in patients with a high risk of ICA injury (malignancy involving the ICA, vasculopathy, coagulopathy, etc.) [11].

The role of intraoperative image guidance in reducing the risk of ICA injury remains unclear. Most studies do not mention using intraoperative guidance; however, it can be assumed that most skull base surgery centers use it routinely [1,22,23].

In the present case study, both a head MRI and CT were performed preoperatively. We routinely use intraoperative imaging guidance in every transnasal endoscopic skull base surgery.

### 3.4. Initial Treatment

ICA injuries can manifest as delayed bleeding or a pseudoaneurysm that can appear anytime, from one day to years after surgery. However, ICA injuries are most frequently identified intraoperatively, as profuse arterial bleeding [24,25]. Initial treatments aim to manage bleeding promptly, before the injury is treated with definitive long-term methods. At the moment of an ICA injury, the most fundamental requirements are cooperation among surgeons, the surgeons’ experience with “four-handed” endoscopic surgery, and cooperation with the anesthesiologist [1,3,6,21].

Direct visualization of the bleeding location is preferred to “blind” sphenoid sinus packing. Overpacking can lead to the compression and complete occlusion of the affected ICA [6,14]. For bleeding management, a binostril approach is considered more favorable than a mononostril approach for resecting the posterior part of the nasal septum. In the binostril approach, the endoscope can be administered through the contralateral nasal meatus, which, combined with the suction–irrigation system, helps maintain good visualization of the affected area. A wide suction tube (4 mm or more in diameter) can be administered through the ipsilateral nasal meatus, and, due to the hydrostatic properties of blood, this suction can help direct the stream towards the ipsilateral meatus, which prevents soiling of the endoscope tip. In cases where a nasoseptal flap was harvested prior to an ICA injury, Valentine et al. recommended using two suction instruments: one to push the flap into the nasopharynx [3].

Gauze strips are most frequently used for primary packing; however, packing with a muscle graft is probably the most reliable method, with superior results than other hemostatic agents [10,11]. This graft is typically harvested from the tensor fasciae latae muscle, and it is crushed prior to placing it on the ICA lesion. Crushing releases calcium from the muscle fibers, which promotes local hemostatic processes [1]. In several cases, initial packing was successful with Merocel, oxycellulose, or Gelfoam [1]. Successful balloon packing of the ICA lesion has also been reported [1,26]. In addition, bipolar coagulation was used successfully to manage bleeding from smaller ICA perforations [1].

Throughout surgery, it is necessary to maintain good cooperation between the surgical team and the anesthesiologist in charge of maintaining stable vital functions. Generally, maintaining normotension is recommended to ensure adequate brain perfusion [8]. However, Fastenberg et al. reported that a bolus of adenosine could be used to induce a short interval of hypotension and bradycardia, which can substantially reduce bleeding, and facilitate the safe placement of muscle graft packing [27].

The present case was treated with a mononostril approach. Therefore, instrument manipulation was difficult in this narrow space. The infiltration of the tumor into the ICA, together with the choice of approach, made it impossible to gain direct visualization of the bleeding location. Therefore, the entire right sphenoid sinus was packed with muscle. Despite prompt crystalloid administration and red blood cell transfusions, the patient remained hemodynamically unstable.

### 3.5. Final Management of the ICA Lesion

In the last decade, packing alone has been considered inadequate for managing bleeding, due to the high risks of rebleeding and delayed pseudoaneurysm development [1,6,25]. Currently, endovascular treatment of ICA lesions during a DSA is considered the “gold standard” method [11].

It is recommended that the balloon occlusion test (BOT) should be performed before endovascular treatment to assess collateral circulation. However, adequate collateral circulation is the not only factor to consider in deciding whether to perform parent artery occlusion or maintain ICA patency (with stent–grafts or flow diverters) during an endovascular treatment, even when the BOT is negative. Indeed, up to 22% of cases developed delayed ipsilateral brain ischemia with permanent neurological sequelae [25]. However, ICA occlusion is a highly reliable technique; Chin et al. showed that bleeding from the ICA was successfully managed with this method in 100% of 46 patients [1]. Therefore, ICA occlusion should be considered when bleeding control is difficult with initial packing.

On the other hand, when not contraindicated, due to a dual antiplatelet regimen or a residual tumor in the affected area, an endovascular treatment that maintains ICA flow should be preferred to ICA occlusion. The patent–ICA endovascular technique had a relatively high success rate, with low residual morbidity: bleeding was successfully managed with no neurological deficit in 58.3% of cases, and minor complications occurred in 33.3% of cases [1,25]. However, when collateral circulation is insufficient, and the patent–ICA endovascular treatment is contraindicated, a craniotomy with an ICA bypass could be considered [28]. Moreover, when the DSA reveals ICA obstruction in a hemodynamically stable patient, the recommended treatment is to loosen the packing to reduce pressure on the ICA and to control the obstruction angiographically [6,15].

In the present case, the DSA revealed complete obstruction of the injured ICA (probably caused by packing pressure on the area affected by the tumor), but collateral circulation was good. Since the patient was hemodynamically unstable, an endovascular intervention was not indicated.

### 3.6. Sequalae

The risk that an ICA injury might cause a neurological deficit is relatively high. The risk depends on the final method of management and the patient’s overall state and disposition. The literature has provided no clear data about the frequency or types of neurological deficits caused by ICA injuries. Undoubtably, this lack of data is due to the fact that many surgeons that encounter this complication do not report it in the scientific literature. In one review, Sylvester et al. showed that a neurological deficit occurred in 19 patients (i.e., 18% of 105 patients) treated with an endovascular intervention. They found that five (5%) of these patients experienced complete resolution of the neurological deficit, nine (8%) experienced partial resolution or a permanent deficit, and five (5%) patients died [25].

The patient–ICA endovascular treatment method can probably provide the best results; it has shown a low complication rate and minimal permanent sequelae. However, very few cases have been reported in the literature. For example, in the Sylvester et al. study, out of 105 patients with ICA injuries, only 9 were treated with this method, and 2 of those patients experienced permanent neurological deficits [25].

The lethality of iatrogenic ICA injury is 10%, according to the literature [1].

In the present case, the patient was hemodynamically unstable and required CPR. This situation resulted in diffuse brain ischemia and malignant edema, which subsequently lead to death.

## 4. Conclusions

An ICA injury is one of the most severe complications that can occur during an endoscopic skull base surgery; it is associated with high morbidity and lethality. According to the literature, the incidence of ICA injuries is low, and few cases have been reported. Most studies have been individual case reports or small case series. To date, there is no clear consensus or guideline for managing ICA injuries. Therefore, an algorithm for managing ICA injuries should be designed for settings specific to each institution where endoscopic skull base surgery is performed.

We lack relevant data on the results of different treatment methods. Based on our literature review, the most promising method is probably initial packing with a muscle graft and a subsequent endovascular intervention. This case study added relevant data to the sparse evidence available on iatrogenic ICA injuries.

The erudition of the surgical team and broad interdisciplinary cooperation among otorhinolaryngologists, neurosurgeons, anesthesiologists, interventional radiologists, neurologists, and rehabilitation teams play fundamental roles in the management of ICA injuries. These important aspects of management lead to the best treatment results.

## Figures and Tables

**Figure 1 brainsci-12-01254-f001:**
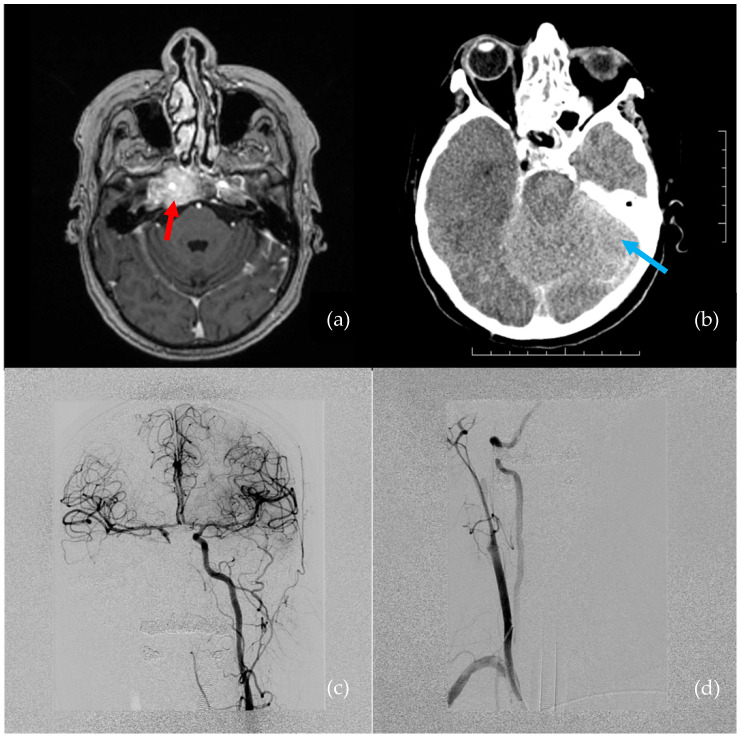
Pre- and postoperative images show iatrogenic injury to the internal carotid artery (ICA). (**a**) Preoperative MRI: tumor involving the right cavernous sinus, encircling the ICA (indicated by red arrow). (**b**) Postoperative brain CT: ischemia and edema in the right hemisphere (indicated by blue arrow). (**c**) Selective left ICA angiogram: good collateral flow through the anterior communicating artery to the right anterior cerebral artery and middle cerebral artery territory. (**d**) Selective right ICA angiogram: occlusion of the cavernous segment of the right ICA.

**Figure 2 brainsci-12-01254-f002:**
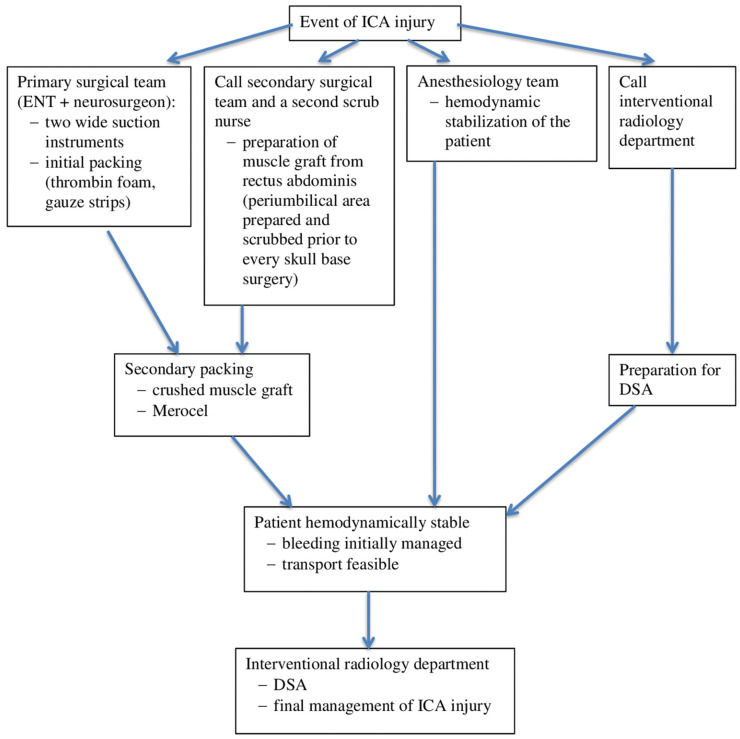
Algorithm for managing an internal carotid artery (ICA) injury, developed at the University Hospital Ostrava. ENT: ear, nose, throat; Merocel: polyvinyl acetate packing strips; DSA: digital subtraction angiography.

## Data Availability

Not applicable.

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
