# Peer review of "Internal Carotid Injury during Skull Base Surgery—Case Report and a Review of the Literature"

_brainsci, 2022, doi:10.3390/brainsci12091254_

Round 1

Reviewer 1 Report

This is a clinical important topic. However, the authors used the formact of a original study to report this case. I think it is not appropriate. Actually, a number of fewer than three cases should be categorized as a case report. Therefore, there is no need to add an additional section of "material and method". Usually for a case report, I would like to break it into (1) introduction (2) case description (3) discussion and (4) conclusion. By the way, in the figures, I would like the authors to point out the main lesions using an arrow or an arrowhead. 

Author Response

Dear reviewer,

Thank you for your kind review and your suggestions. We changed the type of the paper to "case report" and changed its sections accordingly. We also added arrows indicating pathologies into the figures. 

Best regards,

The team of autors

Reviewer 2 Report

Dear Authors,

The present manuscript aimed to definite consensus or guideline for the management of  Iatrogenic injury of the internal carotid artery (ICA).

 I generally liked this topic; I appreciate your effort in this study. I believe this subject is interesting and may lead to clinical implications in the future, however there are some of minor shortcomings that I believe they should improve.

Only, there is some of minor shortcoming,

1. Please write about the detailed method. The method was very general it is better that authors explain about the type of surgical method, the type of local anesthesia, the type of flap in the method part.

2. References are old. I think the authors search again and Revise the references.

3. I did not find any explanation about the search strategy. Please explain about keywords in searching articles.

With Regards

Author Response

Dear reviewer,

thank you for your kind review and your suggestions. Here are our comments:

  1. Please write about the detailed method. The method was very general it is better that authors explain about the type of surgical method, the type of local anesthesia, the type of flap in the method part.

Thank you for this note. The surgical method used in this case was a standard endoscopic mono-nostril approach to the sphenoid sinus; we believe that further description of this procedure would be redundant in case of our paper, since the aim of the report is mainly to discuss management of the complication of the surgery. It was performed under general anesthesia. We added further information about the muscle graft used for packing.

  1. References are old. I think the authors search again and Revise the references. 

Thank you for this suggestion, we searched the literature again and added recent literature to the references.

  1. I did not find any explanation about the search strategy. Please explain about keywords in searching articles.

Thank you for this note. We added information about our search strategy.

Thank you again for your review. Best regards,

The team of authors.

Reviewer 3 Report

1. Firstly, the authors have to change the type of this paper from "article" to "a case report"

2. please to include in the title "a case report"

3. information included in the introduction is insufficient. Please remember that articles read not only by a specialist, but, perhaps, mostly by naive people/scientists. The changes are mandatory.

4. Please better precise why this case is unique.

5. Materiał and methods - please add the anthropometric data, name of company, city, state, country for equipment (CT, MRI). Please add information about the used contrast, its quantity; about sequences used in CT and MRI examinations. 

6. figurę 2 should be replace to the results section and method of screening precisely described in the methods section.

7. I wonder if the figure 2 is your own drawing or you inspired by other authors - please clearly explain it in the main text and add references.

8. Discussion should be focused on obtained results/ presented problem/case.

9. Ethic - please clarify in the method section and also the part "Institutional Review Board Statement" the number of approval, accession date.

10. If (written) informed consent was not applicable, it suggests that presented study was a retrospective study. Please clarify it in whole paper, including the aim of the study.

11. I am not sure if presented case is related to the scope of the Journal. Please consider to cite some papers from this Journal to be sure.

12. In general, the number of references are insufficient. Please also cite more papers from 2021-2022.

13. Conclusion has to be more precise.

14. please make changes in the abstract according to the above comments.

15. English has to be corrected by an English native speaker. Typo mistakes and grammatical errors have to eliminated. For example, in the line 17, the authors should have used the past simple tense instead of the present simple tense (described, not describe) etc.

Author Response

Dear reviewer,

thank you for your kind review and your suggestions. Here are our comments:

Reviewer 3 

1. Firstly, the authors have to change the type of this paper from "article" to "a case report"

2. please to include in the title "a case report"

Thank you for the suggestion, we changed the type and the title of the paper accordingly. 

3. information included in the introduction is insufficient. Please remember that articles read not only by a specialist, but, perhaps, mostly by naive people/scientists. The changes are mandatory.

We extended the introduction section; we believe that now it shows the main aim of the report better; i.e. the proposed algorithm in which the interdisciplinary coordination is crucial. 

4. Please better precise why this case is unique.

We added this information into the introduction - most ICA injuries reported in the literature occured in transsphenoidal surgeries using bi-nostril approach for intrasellar pathologies or in extended approaches to the skull base (i.e. wide corridors, mostly without prior direct involvement of ICA). In our case, the indication for the biopsy was a tumor encircling and invading the ICA and a mono-nostril approach was performed prior to the biopsy - to our best knowledge, both these factors make our case unique in current literature. 

5. Materiał and methods - please add the anthropometric data, name of company, city, state, country for equipment (CT, MRI). Please add information about the used contrast, its quantity; about sequences used in CT and MRI examinations.  

Thank you for this note; in most papers discussing the algorithm of management of ICA injury, mainly from the surgeons' point of view (i.e. papers not focused specifically on the radiologic intervention, or focused on the radiologic aspect of the matter overall) such detailed information about imaging methods is not usually provided, therefore we did not include it either; we believe that it is not substantial for the main aim of the paper, i.e. to propose a management algorithm. However, if necessary from your point of view, we will retrieve the data and include them in the paper. 

6. figurę 2 should be replace to the results section and method of screening precisely described in the methods section.

7. I wonder if the figure 2 is your own drawing or you inspired by other authors - please clearly explain it in the main text and add references.

Because of the changes of the structure of the paper (we changed the "Methods" section to "Case presentation" and merged it with "Results"), we cannot move the figure to the "Methods" section. We believe that its placement in the "Discussion" section is therefore more appropriate, as the algorithm itself it discussed in this section. 

The figure is our own drawing, withou any inspiration by other authors. The title of the figure is " Algorithm for managing an internal carotid artery (ICA) injury, developed at (...our institution...)." which, in our opinion, indicates quite clearly that it is our own work, without any copyright issues or necessity to add any references.  

8. Discussion should be focused on obtained results/ presented problem/case.

The presented case is discussed at the end of every subsection of discussion. We believe that it is appropriate to discuss the topic of each subsection in a broader scope first and the to compare the discussed information to our case. 

9. Ethic - please clarify in the method section and also the part "Institutional Review Board Statement" the number of approval, accession date.

10. If (written) informed consent was not applicable, it suggests that presented study was a retrospective study. Please clarify it in whole paper, including the aim of the study. 

According the the law in the Czech Republic, ethics commitee approval is not necessary for single case reports, therefore it is not applicable.

Prior to surgery, an informed consent was obtained from the patient, including her consent that any images or information, in anonymized, can be used for publication in scientific journals (a copy of this consent was provided to the editor). 

11. I am not sure if presented case is related to the scope of the Journal. Please consider to cite some papers from this Journal to be sure. 

The paper was proposed for publication in the special issue "Improving Outcomes and Preventing Complications in Cranial Base Surgery" of this journal. We believe that our paper is therefore within the scope of this special issue. 

Moreover, an article focused on ICA injury during skull base surgery was published in this journal earlier (Usachev D, Sharipov O, Abdali A, Yakovlev S, Lukshin V, Kutin M, et al. Internal Carotid Artery Injury in Transsphenoidal Surgery: Tenets for Its Avoidance and Refit—A Clinical Study. Brain Sci 2021;11:99. https://doi.org/10.3390/brainsci11010099). 

12. In general, the number of references are insufficient. Please also cite more papers from 2021-2022.

We included more references to current literature. 

13. Conclusion has to be more precise.

Thank you for the comment. We believe that the provided conclusion summarizes the aim of the paper, i.e. to provide a short, general overall view of the current state of knowledge about the ICA injury. We believe that more detailed information and data are discussed in sufficient extent in the discussion section and therefore there is no need to repeat them in the conclusion. 

14. please make changes in the abstract according to the above comments.

The abstract was modified accordingly. 

15. English has to be corrected by an English native speaker. Typo mistakes and grammatical errors have to eliminated. For example, in the line 17, the authors should have used the past simple tense instead of the present simple tense (described, not describe) etc.

The paper was revised by a professional English language editing service (San Francisco Edit). 

Thank you again for your review.

Best regards,

The team of authors.

Round 2

Reviewer 1 Report

The authors have revised according to my suggestion.